# TOWARDS A UNIFIED THEORY OF VALUE AND COGNITIVE MAPS IN THE PREFRONTAL CORTEX

## ABSTRACT

Although the Prefrontal Cortex is known to play a pivotal role in both value-based decision-making and schema learning, the frameworks for each remain largely distinct. By extending recent mechanistic understanding of cognitive maps in PFC to reward-based decision-making tasks, we demonstrate that PFC value responses are necessary for navigating the state-space of the task. Meta-trained RNNs are shown to learn internal value representations that act as control signals for the routing of activity within the network, consistent with the influential Miller & Cohen hypothesis of PFC as an executive controller. This work builds towards a unifying theory of value and schema in PFC, and offers a mechanistic understanding of meta-reinforcement learners, both biological and artificial.

## 1 INTRODUCTION

Two known functions of the Prefrontal Cortex (PFC) are value-based decision-making and schema learning. Over several decades its role in each of these has been meticulously outlined by two influential, yet notably separate, bodies of literature. Both frameworks possess an array of supportive findings across recording techniques such as brain-wide imaging and single-cell electrophysiology, in multiple species from rodents to humans, and causal evidence such as lesion studies, electrical microstimulation and optogenetics. However, although the same sub-regions of PFC are involved in both cognitive functions, we lack a theoretical understanding of how these frameworks might overlap.

The value framework describes the frontal cortex as the accountant of a 'common currency' for decision-making: value (Montague & Berns, 2002; Levy & Glimcher, 2012). Numerous studies have identified neurons across multiple PFC regions (OFC, ACC, LPFC) that code for a range of decision-related variables, such as value of choice (invariant to chosen item identity), value of specific items, chosen option identity, reward magnitude, reward probability, effort required (Padoa-Schioppa & Assad, 2006; 2008; Padoa-Schioppa, 2011; Kennerley et al., 2009; 2011). Frontal value representations are causally necessary for decision-making (Knudsen & Wallis, 2022), as demonstrated by impaired performance in decision-making tasks during optogenetic inhibition of OFC in rats (Ballesta et al., 2020), electrical microstimulation in Macaque monkeys (Gardner et al., 2020), and lesions of human ventromedial frontal cortex (Camille et al., 2011).

The schema framework proposes that frontal cortex learns the structure of experience (Tse et al., 2007; Wikenheiser & Schoenbaum, 2016; Behrens et al., 2018; Niv, 2019; Bein & Niv, 2025; Wilson et al., 2014). These learnt structures, referred to as 'schema' or 'cognitive maps' (Tolman, 1948), are templates that encode the relational or temporal structure that is shared across experiences. For example, in a restaurant there is a typical order of events: ordering, eating, then paying. Human vmPFC and OFC are active during tasks that require inferring position within abstract state-spaces (Constantinescu et al., 2016; Schuck et al., 2016), and OFC neurons have been shown to be tuned to 'position' or 'context' within task-spaces, including on spatial and rule-based tasks (Young & Shapiro, 2011), sequential tasks (Wilson et al., 2014; Samborska et al., 2021; Zhou et al., 2021), and complex hierarchical tasks (El-Gaby et al., 2024).

Recent work has revealed several avenues for bridging between these frameworks. First, a highly influential study modeled PFC as a meta-reinforcement learner (Wang et al., 2018) and recapitulated a suite of behavioral and coarse neural data. However, there is no mechanistic understanding of neural machinery involved in the inner loop of the meta-learner. Second, abstract map-like representations

of value have been discovered in hippocampus (Knudsen & Wallis, 2021). Third, we now have detailed mechanistic understanding of cognitive maps in PFC (Whittington et al., 2025; El-Gaby et al., 2024).

In this work we demonstrate that value signals are consistent with the cognitive map framework when applied to value-based decision-making tasks. In our proposal, value acts as the 'action' which moves the agent within the state-space of available options. To demonstrate this, we meta-train an RNN under both supervised and RL regimes on a value-based decision-making task – a generalized reversal learning task (Samborska et al., 2021). Value representations are observed that act as a control signal for the switching of the choice during reversals. We show that these value representations lie on a line attractor, with reward and no reward acting as velocity signals moving activity up and down the value map. Furthermore, we provide an explanation for why value representations are tuned to time; value representations in the network move dynamically between putative null and potent value subspaces during a trial, which we hypothesise are for temporally orchestrating the switching of the choice at decision time. We believe this new perspective can provide a parsimonious explanation for both value-coding and cognitive maps in PFC, and offer insights into the neural mechanisms inside a meta-reinforcement learner.

## 2 BACKGROUND: STRUCTURED WORKING MEMORY FOR COGNITIVE MAPS

Recent work has provided a mechanistic understanding of how frontal cortex represents cognitive maps (Whittington et al., 2025; El-Gaby et al., 2024). A cognitive map can be formalized as an underlying graph structure populated with observations at each of the nodes (states) (fig1a, top). If an agent understands the structure of the graph—how actions traverse between abstract states—then it can quickly generalise to a novel environment with the same underlying structure but different observations (problem P fig1a).

One such structure is a circular track of 4 states (fig1a). The aim of the agent is to predict the next observation. This should first be possible when returning to the start location for the very first time—despite never having made this transition before. For example, in problem P, after visiting 'apple' for the first time the animal should be able to predict that watermelon comes next. Making this prediction requires leveraging the underlying structure of the task – that state 1 follows state 4.

Both RNNs and PFC solve these tasks by storing all the observations simultaneously in working memory 'slots' (neuron-aligned subspaces that store activity, fig 1a right) (Whittington et al., 2025; El-Gaby et al., 2024). In the 4-loop task there would be 4 working memory slots within the network, for the observations at each of the landmarks. However, rather than learning to structure these memory subspaces by absolute position (state 1, state 2, ...), the network structures memories by *relative* position (now, next, next-next, ..., fig 1a bottom). This requires observations (activity) to move around between the slots as actions are taken in physical space. The transition structure of the cognitive map is encoded in the meta-learnt weights that move activity between the structured slots according to actions. Since the slots can store any observation as persistent activity, this mechanism enables generalization to any 4-loop.

## 3 HYPOTHESIS: STRUCTURED WORKING MEMORY CAN SOLVE VALUE-BASED DECISION-MAKING TASKS

To bridge from map-like tasks to decision making, consider the Harlow task (Harlow, 1949) (fig1b). Each problem contains two random items drawn from a set, with one arbitrarily assigned as rewarded and the other unrewarded. The agent performs multiple trials within each problem, making choices and receiving reward or no reward. After seeing many problems, animals learn the underlying structure: that there is a good option and a bad option, and that a win-stay lose-switch (WSLS) strategy enables inference of the better option after only one trial.

The WSLS solution to the Harlow task can be implemented in a similar manner to the 4-loop track solution. Here, choice and non-choice slots which store the items are analogous to 'now' and 'not now'. The activity in these slots is swapped if there is a 'switch' control signal, analogous to the velocity signal of the 4-loop solution. Reward and no reward correspond to 'stay' and 'switch' signals respectively. As before, this solution abstracts the underlying rule in the recurrent weights

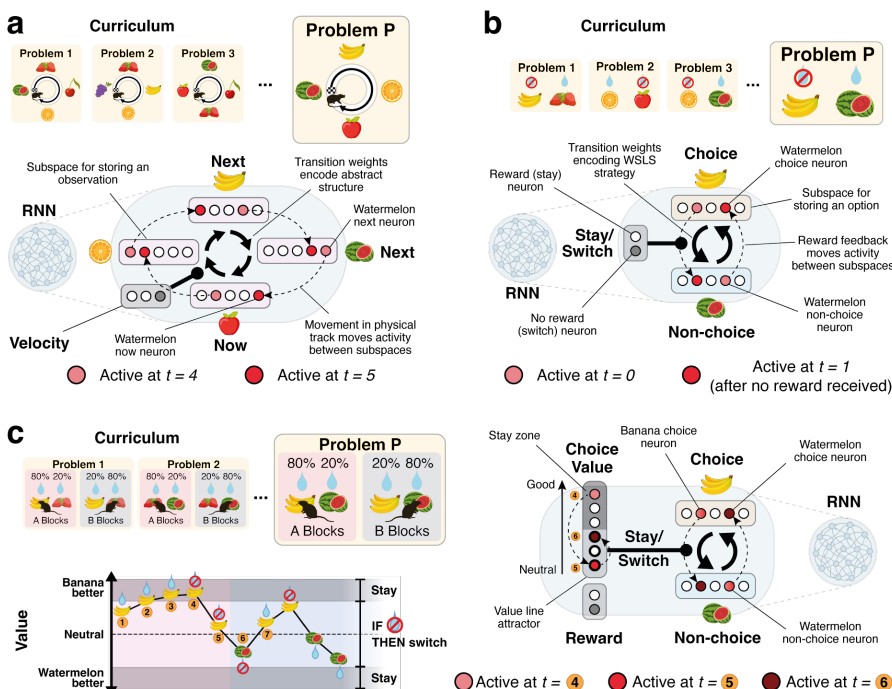

Figure 1: **Neural representations of cognitive maps in PFC**. Each task type consists of a curriculum of tasks with shared structure but different observations. Networks learn to store observations in working memory subspaces called slots, and move activity (dashed lines) between slots according to control signals. Slots denoted as translucent rectangles containing neurons (circles). Control signals denoted with rounded solid lines and transition structure with solid arrows. a) Learnt working memory solution for a spatial 4-loop track. Illustrated at time point $t = 4$, when the agent is at the final state and due to return to the start. b) Hypothetical model for the Harlow task. Illustrated having just chosen banana and receiving no reward. c) Proposed model for reversal learning tasks. Panel illustrates example behavior of a Bayes agent with choices and reward feedback labeled. Shaded grey regions denote values leading to stay, and unshaded regions denote values that can lead to a switch. Network is illustrated at $t \in [4, 5, 6]$ as labeled on the left panel, with a switch occurring at $t = 6$.

that move activity between slots. The slots themselves can store any representation, therefore this mechanism enables generalisation to Harlow tasks containing any objects.

Reversal tasks (fig1c) are an extension of the Harlow task: there is a good option and a bad option, and the agent must infer these contingencies. However, these contingencies reverse according to a block structure and reward feedback is probabilistic, so the agent must integrate a history of rewards into a belief or value. A simple extension to the Harlow solution can support reversal learning (fig1c, right). Rewards must be integrated into a latent belief or value that controls switches. Switches occur only if this value is below a critical threshold and the agent is unrewarded (unshaded region of graph in fig1c left). Therefore computation of the 'switch-stay' control signal requires tracking both value and reward. This is our hypothesis for how RNNs, and perhaps PFC, solve such tasks.

## 4 RESULTS: META-TRAINED RNNS LEARN STRUCTURED WORKING MEMORY SOLUTIONS IN VALUE-BASED DECISION MAKING TASKS

We focus on the generalised reversal learning task (Samborska et al., 2021), but plan to extend the analysis to different tasks in future work. Within each problem there are two available options ($a$, $b$) randomly drawn from a set of $N = 100$ possible options. The reward contingencies for $a$ and $b$ reverse according to a block structure: in $A$ blocks options $a$ and $b$ are rewarded with probability $p_r$ and $1 - p_r$ respectively where $p_r = 0.7$, and vice versa in $B$ blocks. The block reversal probability each trial is $p_s = 1/6$.

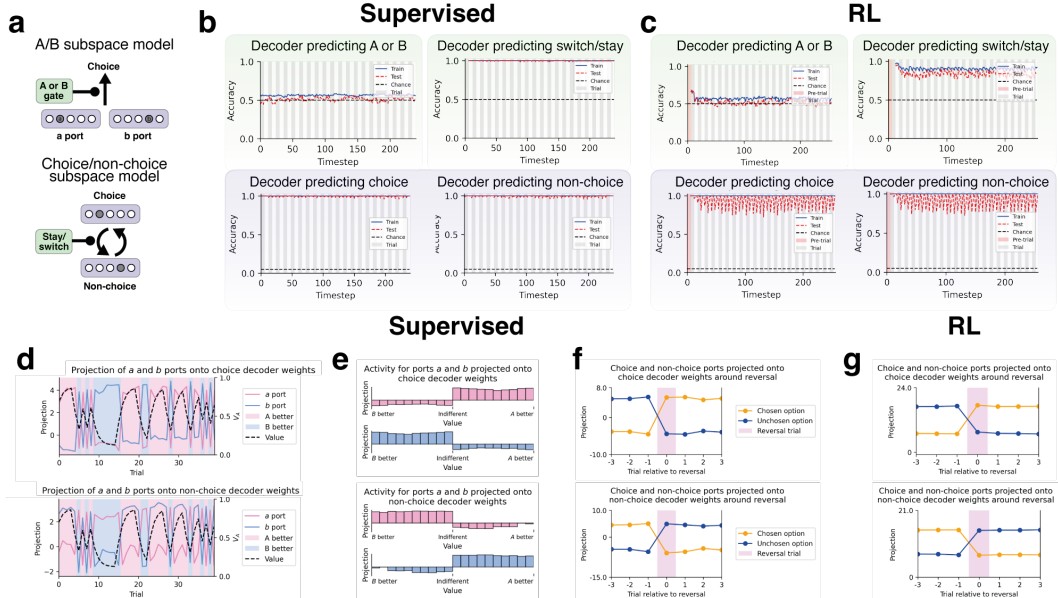

Figure 2: **Choice/non-choice representations in supervised and RL-trained networks**. a) A/B and choice/non-choice hypothetical models. b, c) Decoding of gating signals (top, green) in supervised and RL regimes, and of choice and non-choice options with chance $= 1/N$ (bottom, grey). d) Projection magnitudes of $a$ (pink) and $b$ (blue) options into the choice (top) and non-choice (bottom) subspaces. e) Projection magnitudes of $a$ and $b$ into choice and non-choice subspaces, plotted with respect to value of $A$. f, g) Projection magnitudes of chosen (orange) and unchosen (blue) options into the choice (top) and non-choice (bottom) subspaces in a 7-trial window centred on a reversal (shaded pink), for supervised and RL regimes.

In order to understand how RNNs mechanistically solve these problems in different training regimes, we trained RNNs in both a supervised setting and also a RL setting using PPO[1]. In both settings the RNN was trained on a curriculum of many reversal problems involving different available options $(a, b)$ but with the same $p_r$ and $p_s$ (fig1c, left). For supervised learning we use an ideal Bayes agent as a teacher to generate targets choices for the RNN, and the network consisted of a vanilla ReLU RNN. For the PPO actor-critic regime, the network consisted of separate GRU RNNs for the actor and critic (parameter details in Appendix A.1). All analyses for the actor-critic network were performed on the actor network, meaning that identified value signals were non-trivial.

The relevant options $a, b \in [0, 1, ...N]$ in a problem are presented sequentially as one-hot observations $x_a, x_b \in R^N$ only at the beginning of each problem, and so they must be memorised, in some form, by the RNN for successful behaviour. At each timestep $t$ the previous reward $r_{t-1}$ and action $a_{t-1}$ are provided as input to the network; for the supervised learning regime the teacher's action is provided, whereas for the RL regime the actor's own action is provided. Each problem consisted of 40 trials. For relation to neuroscience data, trials are duration $T$ timesteps with choice required in the middle of the trial and reward of the previous trial $k - 1$ given at the start of trial $k$. In our results we have used different trial durations (of 6, 12, 20) for different models runs, and find results consistent across the different durations.

**Choice and non-choice coding.** As described in fig1c (right) we hypothesised that the RNNs would learn to decompose its representation into distinct subspaces, one that contains the choice option and one that contains the non-choice option. To test this we trained linear decoders from the learned RNN's representation to predict choice and non-choice. Indeed we found that choice and non-choice subspaces could be reliably decoded, and generalised to held out combinations of options (fig2b/c bottom for both supervised and RL networks). This subspace structuring implies the RNN has learned to represent the options in terms of relative position (fig2a bottom), i.e., now/not-now subspaces in which the $a$ and $b$ options move between subspaces, rather than using absolute po-

---

[1]Results for RL using PPO are preliminary and subject to ongoing work.

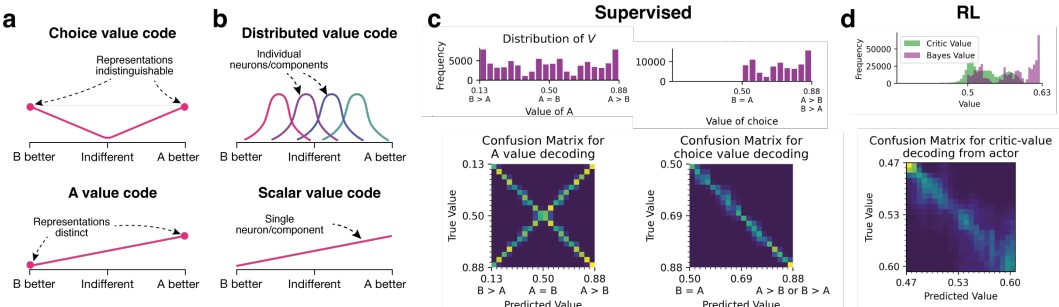

Figure 3: **Value represented as choice-value in both supervised and RL-trained networks**. a) Hypothetical value codes. b) Hypothetical distributed and scalar value codes. Each line denotes the activity of an individual neuron. c) Decoding A value and choice value of the Bayes agent in the supervised network. Note binned-value only decodable with high-dimensional distributed code. d) Decoding critic-value in the *actor* RNN. See Appendix A.1.3 for further details.

sition representation in which the $a$ and $b$ options stay in static subspaces (fig2a top). Indeed we were unable to decode absolute position, i.e., a binary variable saying whether $a$ or $b$ is the chosen option (fig2b/c top left). However, we were able to decode a binary switch (stay) signal (fig2b top-right) which would be required to move (or not move) contents between the choice and non-choice subspaces. These results confirm the (randomly initialised) RNNs trained on RL tasks learn structured subspaces to hold memories of what is currently good (choice) and what is currently bad (non choice).

In both networks, the contents of the choice and non-choice subspaces switch suddenly during behavioural reversals, rather than in a graded manner as a function of value (fig2f,g). Linear decoders were trained to predict choice $c \in [0, 1, ...N]$ (or non-choice) and the beta weights were extracted. The magnitude of the projection the $a$ and $b$ options onto these weights was then used as a proxy for how strongly the option is encoded in the corresponding subspace. The two options exhibit discrete flipping rather than graded movement between the choice and non-choice subspaces, suggesting a discrete trial in which the contents of the good and bad subspaces are switched. We show this by plotting this projection 1) with respect to value of option $A/B$ (fig2e, graded code would be sloped rather than step-like), and 2) relative to reversal trials (fig2f,g, step-like rather than smooth swap around reversal trial).

**Value coding.** In order for a network to know when to swap the contents of the good and bad subspaces it must track value, i.e., when value goes below a threshold then swap the subspaces. After training a decoder for value on the RNN learned representations, we found that choice value, but not $A/B$ value, was linearly decodable from both the supervised RNN and the actor RNN of the actor-critic network (fig3). Choice value is defined as high when either of $A$ or $B$ are good, while $A/B$ value is defined as high when $A$ is good and low when $B$ is good (fig3a). Value was split into 20 bins and logistic regression was used. The 'X' shaped confusion matrix when decoding $A$ value demonstrates that representations of high $A$ value are indistinguishable from high $B$ value (fig3c). Since *binned* value was decodable, it must be represented within the networks in a high-dimensional distributed code rather than a low-dimensional scalar code (fig3b). We note that, in the RL regime (fig3d bottom), we trained the value decoder on the actor RNN (and not the critic RNN) and so the learned value representations must be relevant for taking actions (i.e., swapping the contents of the choice/non-choice subspaces) rather than for computing value for RL itself. Comparison of critic value and the value of an ideal Bayes agent is shown in Appendix A.1.3.

**Switch and stay subspaces.**[2] To execute the switching between choice and non-choice subspaces, we hypothesised the existence of separate subspaces for choice and non-choice in each of 'stay' and 'switch' conditions that are gated on or off by a switch/stay control signal (fig4a). This is just like ring attractors neuron that are conjunctive between position and 'clockwise' or 'counter-clockwise' and are gated on or off by a 'clockwise'/'counter-clockwise' signal.

---

[2]We only consider the supervised setting here as analyses were not ready in the RL setting at time of submission.

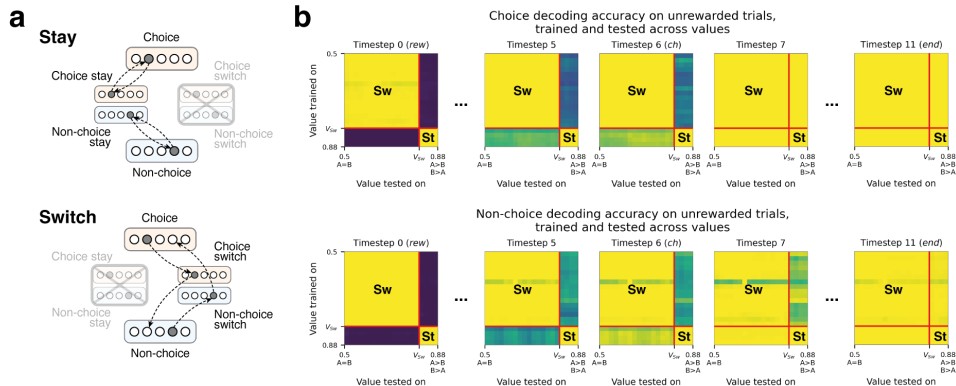

Figure 4: **Switch and stay subspaces for moving activity between choice and non-choice subspaces (supervised network)**. a) Hypothetical model with gateable switch and stay subspaces. Dashed lines denote movement of activity as a result of gating. Stay subspaces projects back to source subspace, while switch subspaces project on to opposite subspace. b) Choice (top) and non-choice (bottom) decoders, each trained on a value bin ($N_{bins} = 20$) and tested across other bins, for timesteps 0 (reward-time), 5, 6 (choice-time), 7 and 11 (trial end). Red line denotes $V_{Sw}$, the value below which switches occur when unrewarded. Block diagonal in early timesteps indicates different choice/non-choice representation for switch and stay conditions.

If these subspaces exist, then a choice (non-choice) decoder trained when the RNNs 'stays' should not generalise to decode choice (non-choice) when the RNN 'switches'. To test for this we first delineate stay and switch trials by finding the specific value, $V_{sw}$, at which if the trial is unrewarded the RNN will reverse its choice. We can then train choice (or non-choice) decoders at specific value bins and assess for generalisation to different values bins (value defined prior to integrating the reward) on unrewarded trials (where switches can occur). This then generates a 2D grid of accuracies as a proxy for subspace similarity (fig4b). For the choice decoders (top), prior to choice-time a block diagonal structured is observed showing decoder generalisation within but not between the switch value range $[0.5, V_{Sw}]$ and stay value range $[V_{Sw}, V_{max}]$. This suggests that there are separate subspaces for 'choice stay' (when value is high and receiving no reward wouldn't reverse your choice) and 'choice switch' (when value is low and so no reward would reverse your choice). After choice time, the choice decoders generalise across values, implying that the chosen option has moved into the choice subspace. In contrast, the non-choice decoder plot stays block diagonal for longer, suggesting that the non-choice option takes additional time (within a trial) to move to the non-choice subspace.

**Value representations form a line attractor.** In order to update value appropriately, we hypothesise that value in represented by a line attractor. We have already shown that value code is distributed since binned value be linearly decoded (fig3). However in order to be a line attractor there must be the appropriate mechanisms to move the value up or down in value. This requires separate gateable populations for moving value activity up or down subject to reward or no reward (analogous to clockwise/anti-clockwise head direction populations in the fly ring attractor or the stay/switch mechanism in the previous section) (fig5a). As we predict, a decoder trained to predict previous value on rewarded trials does not generalise to unrewarded trials and vice-versa (fig5b), indicating the presence of rewarded-value and unrewarded-value subspaces.

To visualise the value line attractor, PCA was performed on an array containing the mean hidden activity for each combination of previous-value bin and rewarded/no rewarded feedback (fig5c). We then projected the activity (with reward regressed out) on the principal components to see how value is represented and how it differs in rewarded and unrewarded conditions. We first isolate two principal components, PC0 and PC2, in which activity spans axes representing graded value. We observe that, in early timepoints the axes span previous value in a continuous manner but separated by reward; this is necessary for updating value in the correct direction, and for deriving the switch signal (low value & unrewarded). In middle timesteps (2-3), we observe the the two value lines are entirely separated confirming the two value subspaces (one active when rewarded, the other active when unrewarded; note the separation cannot be explained by a reward input since we regressed out

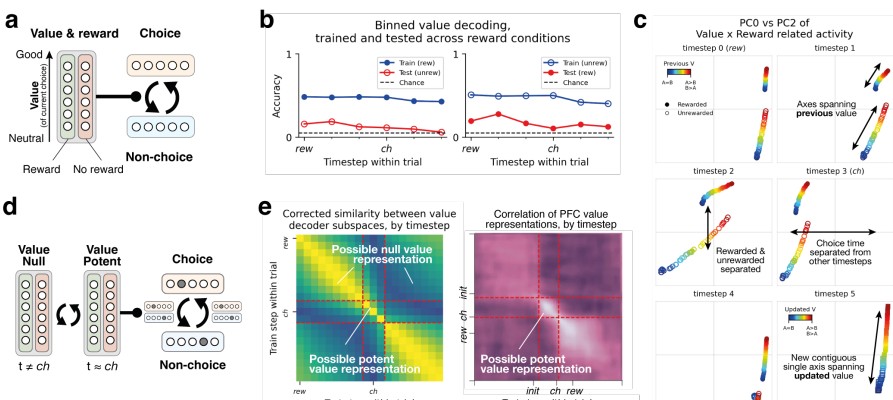

Figure 5: **Value representations form an attractor and move through time (supervised network)**. a) Proposed model for value attractor with rewarded and unrewarded choice value axes. b) Value decoder performance when trained on rewarded trials (left) and on unrewarded trials (right). c) PCA on concatenated mean activity (having regressed out reward) across all combinations of reward and value. Separation with respect to reward is observed. Note, final plot colour-coded by updated value. d) Hypothetical model with null and potent value subspaces, with the potent subspace controlling switch/stay gating. e) Left: cosine similarity analysis of decoder beta weights between decoders trained on different timesteps. Right: equivalent analysis from PFC data showing correlations between regression weights of a policy encoder (Samborska et al., 2021)

reward). In the final subplot we colour by the updated value and show a single contiguous value axis.

**Value representations move through time.** In the proposed model the decision to switch occurs only once per trial and at a specific time point. Analogous to the output-null and output-potent subspaces for executing actions in motor control (Kaufman et al., 2014), we hypothesise the existence of null and potent value subspaces that control the timepoint at which the contents of the choice and non-choice subspaces are switched (by gating the switch and stay subspaces). We anticipate the value subspaces are null and potent because a low value signal in conjunction with being unrewarded (which is an existing subspace as per the previous section) is the requisite signal to switch.

To test for this, we trained value decoders on each timestep of a trial and tested across all others, to assess subspace re-use. However, since the internal update of the representation of value from the previous trial $V_{k-1}$ to value of the current trial $V_k$ occurs at some unknown timepoint after reward feedback, naive decoder generalisation will be poor across time within the trial. To account for this we devised a value-subspace similarity analysis (fig5e) (Appendix A.2.2) that assumes values are updated according to their standard transition statistics. We observe that a more similar value subspace is used before and after choice time, while the value representation around choice time is distinct (red dashed lines). This suggests a different value subspace is active during choice time as compared to other timepoints. This result mirrors findings from the study on which our task is based (Samborska et al., 2021), which found policy representations that shift depending on the chunk of the trial, with a distinct block just prior to choice-time.

This results additionally explain why in the PCA plot (fig5c), value is only updated towards the end of the trial, even though all the information to update value is present at the trial's beginning when reward is provided. In particular, in order to have a switch/stay signal—low previous value (i.e., un-updated value) in conjunction with unrewarded—available up until choice time (which is when the switch occurs), value cannot be updated. Instead value must wait until after the potent subspace has been active and only then can it update.

**Single-cell tunings elucidate potential gating mechanism.** We now present single cells that correspond to the various components of the choice/non-choice circuit identified above.

*Choice cells*: Cells tuned to specific choices (cell tuned to choosing options 1 and 12 fig6a) which are invariant to value, firing whenever the preferred option is the choice (i.e. is the better option). These

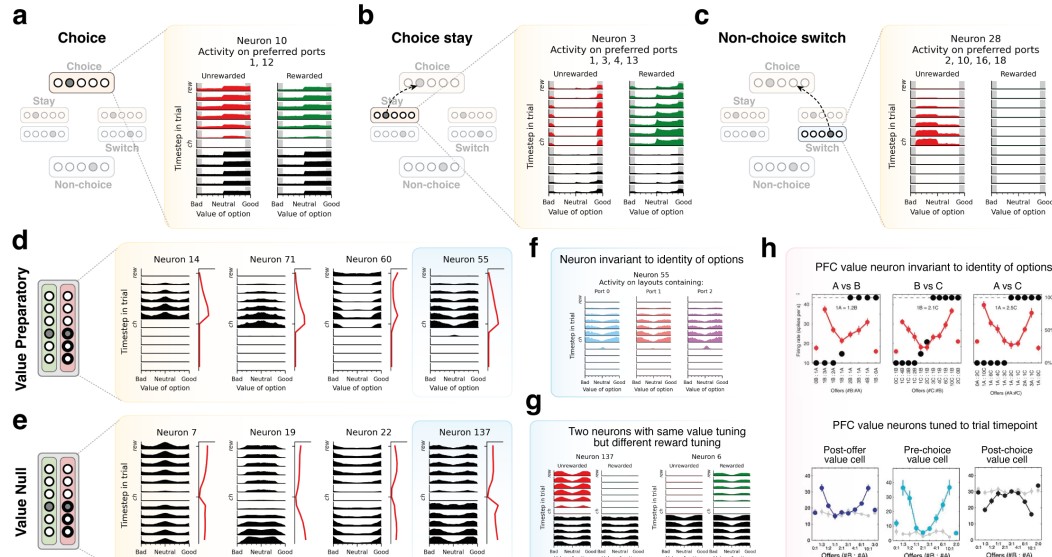

Figure 6: **Example single cell tunings (supervised network)**. From a simulation with 12 timesteps. Choice-time $ch$ and reward-time denoted $rew$. In neuron plots, x axis corresponds to value spanning bad-neutral-good. **a)** Choice neuron with preferred options 1, 12 and is invariant to reward. Shaded grey regions denote values above value threshold for switching, $V_{Sw}$. **b)** Choice-stay neuron, tuned to options 1, 3, 4, 13. Convention as in a). Neuron fires if preferred option is good and decision is to stay. **c)** Non-choice switch neuron. Convention as in a). Neuron active if preferred option was bad (non-choice) and decision is to switch. **d)** Value preparatory cells. Each panel shows the ratemap on the left and the mean temporal profile on the right. Two neurons highlighted blue and showcased to the right. **e)** As in e) but for Value null subspace. **f)** Example value cell that is invariant to which options are included in the problem. **g)** Two example value cells with tuning to reward (right cell) and no reward (left cell). **h)** Example tuning curves of PFC value cells. Top shows value cell that is invariant to the identities of the items on offer. Bottom shows three cells tuned to different timepoints of the trial (Padoa-Schioppa & Assad, 2006; 2008).

cells are typically active at all timepoints other than choice-time, suggesting they act as working memory storage of the choice.

*Switch and stay cells*: Two cells tuned to choice-stay and non-choice-switch are shown (fig6b, c). The first cell fires only for preferred options 1, 3, 4, 13 under stay conditions: when rewarded, or when the value was sufficiently high and unrewarded (grey region corresponds to region in fig1c graph). The second cell is tuned to options 2, 10, 16, 18 under switch conditions: low value and unrewarded. This cell is non-choice switch, since it is tuned to the bad option.

*Value cells*: Several value-tuned cells (fig6d, e) that display symmetric tuning curves (fire identically if an option is equally good or bad, as implied by the 'X' confusion matrix from fig3b). Value cells are invariant to the specific options included in the problem (an example cell fires identically in problems containing options 0, 1 or 2, fig6f). Some value cells are also co-tuned to reward or no reward (fig6g), in accordance with the attractor model which requires separate populations for moving 'up' (rewarded) and 'down' (unrewarded) the value axis. This tuning disappears after choice-time, as value partitioned by reward is no longer needed to compute the switch signal. The tuning curves are visually similar to those observed in OFC during economic decision-making tasks (Padoa-Schioppa & Assad, 2006; 2008) (fig6h), both in terms of shape and the time-selectivity.

*Value cell time tuning consistent with population dynamics*: Value cells were tuned to time within a trial; two broad categories were identified, which we refer to as 'Value Preparatory' and 'Value Null'. Value preparatory neurons exhibit ramping activity until choice-time (fig5d). Their hypothesised role is to make the switch/stay decision in anticipation of choice-time, by gating switch/stay subspaces on or off (fig7a,c pre choice-time & choice-time), and are inactive after choice-time as the decision has been made (fig6d and fig7b). The time-courses of the preparatory and null sub-

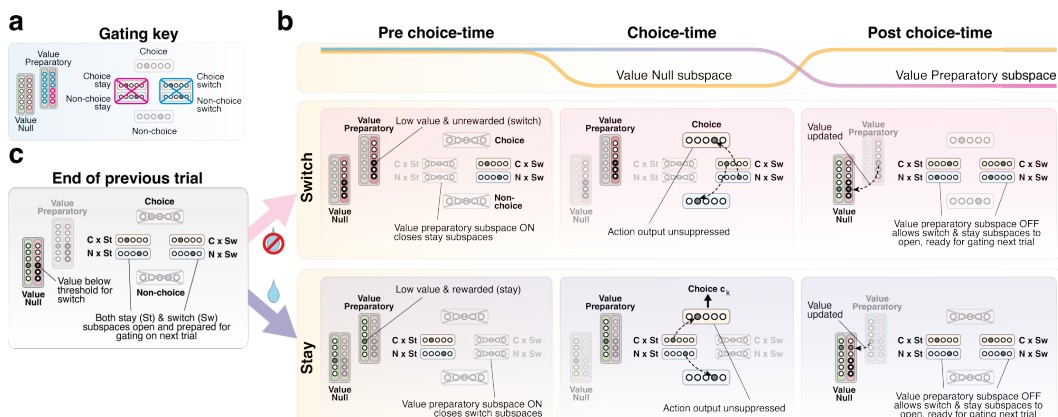

Figure 7: **Putative gating mechanism**. **a)** Gating scheme, pink denotes switch (stay gated off) and blue stay (switch gated off). **b, c)** An example trial involving either switch (no reward, top) or stay (reward, bottom). Translucent denote inactive subspaces, opaque denotes active. Value update occurs when activity passed back to null post choice-time.

spaces are consistent with the cosine similarity analysis in fig5g, which showed maximal difference in value representations around choice-time, when preparatory activity is maximal and null activity is minimal. Gating of switch and stay subspaces by value preparatory neurons prior to, but not after, choice time is consistent with fig4.

## 5 DISCUSSION

We demonstrated that RNNs (in a supervised setting and with preliminary results using RL) trained on a generalised reversal task learn to 1) store memories of the 2 available options in choice and non-choice working memory subspaces, 2) integrate reward into value using a line attractor, 3) gate movement of activity between choice and non-choice subspaces in order to change preference, 4) orchestrate gating at a specific time point within each trial using null and preparatory value subspaces.

These findings are consistent with evidence from PFC literature: 1) options are represented in the choice/non-choice frame of reference (Padoa-Schioppa & Assad, 2006), 2) rapid switches between choice preferences on reversal tasks, consistent with inference (Bartolo & Averbeck, 2020) rather than RL, 3) value as a map which is navigated by reward, similar to that found in hippocampus (Knudsen & Wallis, 2021), 4) time tuning in value cells, as observed in all PFC recordings of value-tuned cells (Padoa-Schioppa & Assad, 2006; Padoa-Schioppa, 2011; Kennerley et al., 2009; 2011).

In fig7 we summarise our overall proposed mechanism of how structured working memory representations can perform effective RL in value-based decision making tasks. While there are some aspects of this algorithm that we have not fully proved, we anticipate it will be of broad relevance to mechanistically understanding RNNs and PFC in decision making tasks. In order to prove this circuit conclusively there are several avenues to explore. First, we must prove value subspaces perform gating operations on the choice/non-choice circuit (e.g., by effective connectivity between subspaces or causal manipulations). Second, we must replicate all supervised results with RL (rather than the preliminary results we present here). Third, we must show that conceptually the same circuit is used for other value-based decision-making tasks such as uncorrelated bandits, one-shot neuroeconomics decision-making tasks, two-step model-based tasks.

We believe the contribution of this work is two-fold: 1) we expand upon a highly influential work in which PFC is modelled as a meta-reinforcement learner (Wang et al., 2018) by providing a mechanistic understanding of the neural representations and algorithms learnt within such networks, 2) we demonstrate that theoretical ideas from the cognitive map framework can be extended to the value-based decision-making framework. Value as an internal gating signal is a departure from the conventional interpretation, in which value is merely an output sent by PFC to Striatum, and brings PFC value representations in line with the Miller & Cohen framing of frontal cortex as an executive controller (Miller & Cohen, 2001) that gates the flow of information around the brain.

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

# A APPENDIX

## A.1 MODEL & TASK IMPLEMENTATION DETAILS

### A.1.1 EPISODE STRUCTURE

Episodes consist of an episode setup phase, in which the relevant options in a problem $(a, b)$ are presented sequentially as observations. For the remainder of the episode at each timestep the null observation, previous action and previous reward are provided to the embedding layer.

Table 1: Problem structure

| Parameter | Value |
|---|---|
| Episode setup duration | 6 |
| Trial length | 6 |
| Rollout length | 256 |
| Reward probability $p_r$ | 0.7 |
| Reversal probability $p_s$ | 1/6 |
| Min block length | 1 |

Table 2: Supervised learning parameters

| Parameter | Value |
|---|---|
| Batch size | 128 |
| Learning rate | 1e-4 |
| Optimiser | AdamW |
| L2 activity reg | 1e-4 |
| L2 weight reg | 1e-6 |

### A.1.2 SUPERVISED LEARNING

**Learning algorithm**

Cross-entropy loss on actions provided by teacher (ideal Bayes agent). Initial action is randomly selected between $a$ and $b$.

**Network**

RNN receives concatenated input (observation, action, reward) at each timestep and outputs action logits. Actions on the environment and provided as input to the network are those of the Bayes teacher.

Table 3: Supervised network parameters

| Parameter | Value |
|---|---|
| $N_{options}$ | 100 |
| Hidden dim | 256 |
| Cell type | Simple |
| Activation | ReLU |
| Layer Norm | None |
| Hidden weight init | Identity + $\mathcal{N}(0, 0.001)$ |
| Input weight init | Xavier Uniform(gain=1.0) |
| Output weight init | Uniform(-k, k) where $k = sqrt($input dim$= 202)$ |
| Bias init | 0.0 |

For visualising the single cells in figure 6, networks were trained with L2 regularisation on hidden activity and weights to encourage learning of neuron-aligned subspaces (Whittington et al., 2022)

### A.1.3 REINFORCEMENT LEARNING

**Learning algorithm**

Meta-RL experiments were done with PPO algorithm. Deltas and advantages were not allowed to propagate between trials, since trial reward is independent of actions in previous trials. Hungar-

ian loss was used to encourage the network to retain the relevant options, without restrictions on permutation:

$$L_H = min(L_{CE}(a, y_1) + L_{CE}(b, y_2) , \ L_{CE}(a, y_2) + L_{CE}(b, y_1))$$

Where $L_{CE}$ is cross-entropy loss, $y_1, \ y_2$ are linear logit readouts from the actor RNN, and $a, \ b$ are the identities of the relevant options in a problem.

Table 4: PPO parameters

| Parameter | Value |
|---|---|
| Batch size | 1024 |
| Max grad norm | 0.5 |
| Learning rate | 1e-3 |
| Optimiser | Adam |
| Epochs per update | 4 |
| Minibatches per epoch | 4 |
| Discount factor $\gamma$ | 1.0 |
| $\lambda_{GAE}$ | 1.0 |
| PPO clipping $\epsilon_{clip}$ | 0.2 |
| Entropy loss coef | 0.01 |
| Value loss coef | 0.5 |
| Hungarian loss coef | 0.1 |

**Network**

Networks were sequentially composed of 1) a shared embedding layer for transforming observation, action and reward into input, 2) a separate RNN for each of actor and critic, 3) linear readout heads for: value from critic RNN, policy from actor RNN, logits corresponding to the two options in the problem (Hungarian algorithm detailed above). Hungarian algorithm is only applied to the actor network, though this has not yet been explored further.

Table 5: Actor-critic network parameters

| Parameter | Value |
|---|---|
| $N_{options}$ | 50 |
| Embedding dim | 128 |
| Actor hidden dim | 256 |
| Critic hidden dim | 256 |
| Cell type | GRU |
| Activation | ReLU |
| Layer Norm | Pre-activation |
| Embedding weight init | Orthogonal(0.01) |
| Hidden weight init | Identity $+ \mathcal{N}(0, 0.001)$ |
| Readout heads | Orthogonal(0.01) |

**Consistency with Bayes ideal agent**

Both shared and split actor-critic networks exhibit the V-shape that would be expected when comparing value to Bayes belief, though performance is better with a shared RNN. Top right horizontal histogram shows the critic value distribution (green) overlaid on the Bayes value distribution (blue):

$$V_{Bayes} = p_r \times b_{rect} + (1 - p_r) \times (1 - b_{rect})$$

where reward probability $p_r = 0.7$, and $b_{rect}$ denotes rectified Bayes belief that A is better:

$$b_{rect} = |b_A - 0.5| + 0.5$$

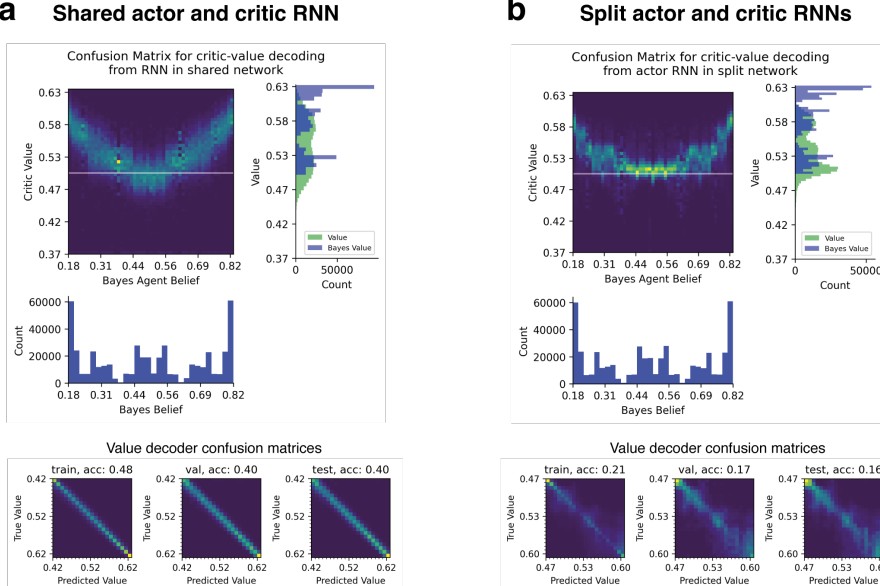

Figure 8: **Critic performance in PPO-trained networks with shared (a) and split (b) actor-critic RNNs.**

## A.2 ANALYSES

### A.2.1 REGRESSION

Analyses were performed on the subset of layouts containing options 1-20. Decoder train set consisted of layouts sampled from the model training set, and test sets from the model test set. For all regression analyses we used the Logistic Regression module from sklearn, with regularisation C=1.

### A.2.2 COSINE SIMILARITY ANAYLSIS FOR VALUE SUBSPACES

We trained separate decoders to predict value $V_k$ of the current trial $k$, at each timepoint $t$ of the trial (9a). However, this did not generalise across time due to the update in value after receiving reward on the first timestep of the trial. We did not know the exact time that the network processes this update and therefore developed this analysis that is ambivalent to the update time.

The beta weights of each $V_k$ decoder trained on a timestep were compared with the beta weights of decoders trained on all other timesteps. Each comparison between decoder $i$ and decoder $j$ yields a 2D RSA matrix (9b, smaller red-blue inset plots) containing the representational similarity for each value in the two decoders. A decoder compared with itself yields an approximate identity matrix (fig 9b top left). A decoder trained on the final timestep and compared with a decoder trained on the first timestep yields an RSA (fig 9b bottom left) that is identical to the transition matrix $M$ for value (fig9b bottom right). This indicates that value has undergone an update.

For each $i, j$ comparison we took the raw RSA and the RSAs when transformed according to $M$ and $M^{-1}$. Between these three RSAs, we identified that with the strongest mean diagonal, and used this to generate a composite RSA (fig9d, uncorrected is shown in fig 9c i.e. mean diagonals of fig9b).

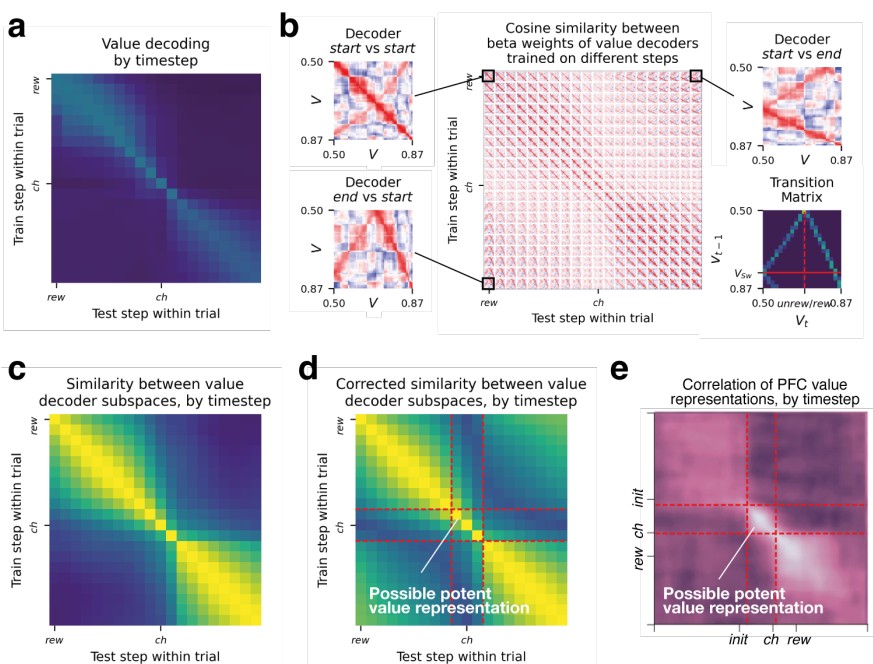

Figure 9: **Cosine similarity analysis to identify value-time representations**. a) Accuracy of value decoders trained and tested across timesteps. **b)** Cosine similarity analysis of decoder beta weights between decoders trained on different timesteps. Each comparison is a small block demarcated by white lines. Red denotes high correlation (1), white no correlation (0), and blue anti-correlation (-1). Three specific comparisons are shown, value decoder at start of trial compared with itself (top left), value decoder at start compared with end (top right), and value decoder at end compared with start (bottom left). Transition matrix for value from the previous trial $V_{k-1}$ to the current trial $V_k$, for comparison. Red solid line denotes the threshold value $V_{Sw}$ below which switches can occur, and red dashed line denotes the mapping from $V_{k-1}$ to $V_k$ depending on reward (left) and no reward (right). **c)** Mean diagonal of each block from **b**, yellow denotes 1, blue denotes 0. **d)** Same as **c** but after correcting cosine similarity in **b** for the value update due to the transition matrix. Red-dashed lines denote putative value representation around choice-time that differs from all other timesteps. **e)** Equivalent analysis to **f** from PFC data Samborska et al. (2021). Value regressed onto neural activity at each timepoint and then regression weights were correlated across time.

