# OpenReview forum: "Towards a unified a theory of value and cognitive maps in the Prefrontal Cortex"
_ICLR.cc/2026/Conference — ICLR 2026 Conference Withdrawn Submission_

### Official Review · Reviewer_1oaZ · 2025-10-31

**Soundness:** 2
**Presentation:** 3
**Contribution:** 2
**Rating:** 2
**Confidence:** 4

**Summary:**

The manuscript attempts to unify the cognitive map theory with mechanisms underlying navigation through value functions. The authors train recurrent neural networks (RNNs), using either supervised or reinforcement learning (RL) approaches, to solve the generalized reversal learning task. The trained RNNs develop choice, value representations, and attractor dynamics that recapitulate experimental data. The work aims to mechanistically connect structured working memory representations, value tracking, and action switching within a unified neural framework.

**Strengths:**

- The authors successfully demonstrate that their RNN model develops attractor dynamics and single-cell representations strikingly similar to those observed in experimental literature, supporting the biological plausibility of the model.

- They attempt to verify the generality of their results by implementing both supervised and RL-based training, though this dual analysis appears as a work in progress.

- The manuscript is well-illustrated, featuring conceptual and explanatory figures that clarify both the logic of the tasks and the key computational mechanisms learned by the network.

- The proposal presents an interesting synthesis of ideas from cognitive mapping, value representations, and neural network dynamics, and frames new hypotheses for how such structures could emerge in biological systems.

**Weaknesses:**

- Scope of Task and Analysis: The study focuses solely on a single task (generalized reversal learning), which limits the generalizability of its claims. Moreover, similar tasks have been previously explored using simpler models (e.g., Zhang et al., 2018, with supervised reservoir networks). Much of the analysis concentrates on the supervised learning setup, as the RL experiments are still preliminary. This split risks conflating results; the manuscript would be clearer if it focused on a single, fully analyzed regime. The added value of analyzing both SL and RL settings in parallel needs clearer justification. What insights does the RL framework bring, and do we expect significant differences?

- Insufficient Coverage of Prior Work: The intro and discussion omits relevant literature connecting TD-error-driven learning and working memory gating mechanisms in both tabular (O’Reilly & Frank, 2006; Lloyd et al., 2012) and neural network models (e.g., Kumar et al., 2021). The latter specifically examines how the TD error drives working memory gating rule/schema learning and should be discussed as it touches directly on the manuscript’s core questions.

- Role and Necessity of RNNs: The authors argue that tracking both value and reward is critical for computing the switch-stay control signal. However, in MDPs lacking temporal dependencies (i.e., states do not depend on the past), such computations should be achievable with a feedforward network. The necessity of the RNN becomes apparent only for problems with temporal dependence (POMDP structure). Distinguishing these settings is important, as is clarifying why reversal learning is a sufficiently challenging or general testbed. Including a simple feedforward baseline or designing tasks with less working memory demand would help clarify what unique computations or representations are enabled by recurrent structure, and what other minimal computations are needed to traverse the cognitive map.

- Claims of Line Attractor Dynamics: While the visualizations suggest line-attractor behavior, the manuscript does not rigorously verify this. Proper demonstration requires analyzing the RNN Jacobian along the attractor and showing (i) a (near-)unit eigenvalue for slow dynamics along the attractor and (ii) all other eigenvalues with magnitude less than one for transverse stability. Without this, the robustness of the inferred attractor in the face of perturbations is unclear.

- Robustness and reproducibility: How many random initializations (seeds) were used for each network configuration? Were the observed proportions of cell types (choice, stay, switch, value) consistent across seeds (e.g., as in Figure 6)? A single-seed analysis may not be representative or robust.

**Questions:**

- The authors state that “when value goes below a threshold then swap the subspaces.” Is the switch ultimately driven by value representations, or by a TD-error-like signal? How do the two relate?

- Figure 2 reports similar decoder accuracy and port selections for both supervised and RL-trained networks. What distinct insights, if any, are gained from these two training paradigms? In particular, how do value representations (as in Figure 3d) diverge or align across regimes?

- On the necessity of recurrence: Could the schema-learning results be replicated by training only the readout gating policy, as in Kumar et al. (2021)? Can the authors discuss the role of RNN dynamics versus simpler architectures in learning the relevant computations?

- Wang et al. (2018) showed that discrete flipping of network dynamics in bandit tasks can arise purely from reward input signals. Here, the reversal task also induces such switches. To what extent is this a generic consequence of reward-driven gating, and what new mechanistic insights does the current work provide?

References:
Zhang et al. 2018 (https://doi.org/10.1371/journal.pcbi.1005925)
O’Reilly & Frank 2006 (https://doi.org/10.1162/089976606775093909)
Lloyd et al. 2012 (https://doi.org/10.3389/fncom.2012.00087)
Kumar et al. 2021 (https://arxiv.org/abs/2106.03580)

---

### Official Review · Reviewer_mZS9 · 2025-10-31

**Soundness:** 2
**Presentation:** 3
**Contribution:** 2
**Rating:** 4
**Confidence:** 4

**Summary:**

Multitask learning RNN setup, trained on a family of value based cognitive tasks with a win stay switch lose strategy via metalearning. The network shows aspects value encoding reported for brain responses in areas of PFC, while recapitulating a very minimalistic form of the cognitive map type of behavior reported previously for the same model with more spatial tasks in (Whittington et al., 2025; El-Gaby et al.,
2024). The claim is that the model is able to reconcile the two sets of empirical observations as state transitions under an action space defined by immediate reward.

**Strengths:**

Modeling of learning across family of tasks and its circuit consequences is important and with relatively few success stories to date. Adding to that set is of definite value to the community.

**Weaknesses:**

I just feel that the overall analogy of reward as action is lose and a little superficial.

Unclear how to generalize beyond the two discrete states state space and lose, win 2d actions to more general problems. Calling a switch between two states a cognitive map i also find a bit of a stretch, which somehow reduces the value of the claimed unification.

Given that the model is essentially that in past work (Whittington et al., 2025; El-Gaby et al., 2024) applied to a different problem, the innovation needs to be judged by the strength of the application in terms of match to biology. It is not clear to me that the results are general enough and substantial enough for the paper to get a strong score by that metric.

Several claims are made about the nature of the resulting representation that are only weakly if at all supported by the presented analysis.

The writing is not particularly clear; the exact setup could have been described better, in particular in relation to past work and what is unique to this paper.

**Questions:**

I found figure 1 very difficult to parse and the background text only clarifies it partially. There are 4 states but what is the action space? go left and right? under what policy? what do you mean exactly by 'meta-learned weights'? I would hope to get the gist of the core idea from the text alone without having to read several past papers...

Am i correct that all tasks considered have only 2 states and the action space is by construction defined to be getting or not getting a binary reward? I am having a hard time calling a 2 state system a 'cognitive map' Can you provide concrete examples (conceptual not numerical) of versions of this idea that have more than 2 states ?

what was the reason for presenting the inputs only at the beginning? it's not clear to me that this is the case for the actual experiment, please justify

given that there are only 2 possible options why is the decoding of choice or non-choice leading distinct subspaces?
i must have missed something but if probability of reward received and switching are constant overall, do different objects correspond to different reward magnitudes and how is that set up?

is the value of reward for each object the 'observation' that gets attached to each state? how is value of an offer encoded ? probably related to the statement of one hot observations being in R^N... is the identity of the channel signalling which object and the nonzero real number its associated value somehow? or the agent gets to learn which channel is on first and then the reward associated to that at the end of the trial?

can you show the low d projection of the actual activity in the task? might make it less opaque compared to the decoding results summary. also the hypotheses about the nature of representation (ring attractor, distributed codes) should be validated more directly rather than just being intuited from the decoding results. the strength of evidence of those claims seems too weak to me.

Similarly, activity moving along a one dimensional subspace within PC axes is not enough to claim line attractor status of the representation: have you tried extracting slow points numerically by kinetic energy minimizations and/or linearizing the dynamics around the slow manifold to prove some local properties of the flow are consistent with that hypothesis?

Treating rewards received as observations and building discrete internal beliefs about the context sounds similar to the value based decision making task in the Constantinople lab, which has also been modeled in a metaRL type of setup (Hocker et al, 2025). I wonder if you could comment on any relation between your network representation and that work, especially since they have more than 2 discrete states...

I found fig6 hard to see and hard to parse. there must be a more compact way to show example neurons encoding and a way to do summary statistics of their overall properties if that's what is intended.

FIgure 7 is interesting as a hypothesis but weakly supported by data and elements too tiny to see properly (general problem with all figures imo)

Overall, some interesting ideas but the results seem still a little preliminary and not completely worked out.

---

### Official Review · Reviewer_7xrQ · 2025-11-01

**Soundness:** 3
**Presentation:** 1
**Contribution:** 3
**Rating:** 6
**Confidence:** 4

**Summary:**

This paper seeks to explain disparate findings in prefrontal cortex (PFC) through a unified model. Specifically, they link  value-based decision-making and cognitive map learning by suggesting that value signals can act as internal control signals to transition through the cognitive map. The authors meta-trained recurrent neural networks on a generalized reversal learning task with both supervised and reinforcement learning. They find that networks learn to represent options in structured choice and non-choice working memory subspaces that swap contents during reversals, represent value in a line attractor, and can gate activity between choice/non-choice subspaces. They discuss how using value as an internal gating signal (as opposed to just an output signal) is a new interpretation of PFC's role.

**Strengths:**

- The topic is of great interest to neuroscientists, and the work seeks to explain and connect prior findings in both experiments and theory.
- The experiments conducted seem thorough and authors use a variety of analyses from the population level to single cell level.

**Weaknesses:**

- I think the greatest thing this paper could improve is presentation and clarity. I found it rather dense and hard to follow.
- The RL regime is more realistic than the supervised regime, but only is used in some of the analyses. It's unclear how much we should expect the results from the supervised regime to generalize to the RL regime, and some discussion about that would be helpful.
- I actually think the slot analogy confuses the point of the paper and makes it harder to understand. Ultimately, the idea in the reversal task is fairly simple-- there's a subspace for choice and a subspace for non-choice; things may move around between the subspaces.
-Also, given that the task actually used in the paper is the reversal task, spending a lot of time talking about the multi-loop task and the relative position slots (now/next/next-next) adds on to the difficulty of reading the paper. It just feels like the introductory explanation could be simplified more to its core points.

**Questions:**

- In Fig 1A, I found it confusing that some sensory observations remained the same across problems. Just feels like it adds unnecessary confusion. Why not make them all novel? Also I don't know why one can zero-shot infer watermelon after the first time you see apple-- I thought the sensory observations were unrelated from loop to loop. Overall, I found the explanations in the section "Background: Structured working memory for cognitive maps" to be hard to understand. The discussion about activity moving between slots was especially confusing. I also think it would be a little better for clarity if there was a disambiguating number between loops and landmarks. Right now, there's 4 loops and 4 landmarks. Would be nice if just one of those numbers is different so this sections is easier to follow.

---

### Official Review · Reviewer_j9iB · 2025-11-01

**Soundness:** 1
**Presentation:** 3
**Contribution:** 1
**Rating:** 2
**Confidence:** 4

**Summary:**

This paper demonstrated that when a recursive neural network (RNN) is trained on a well-known task in the neuroscience literature, i.e. generalized reversal learning, some of its behavior and components mimic the representations and observations from the brain, especially in the prefrontal cortex and the hippocampus. The authors claimed that this is an evidence in favor of theories that connect the value representation and cognitive maps for decision making. While the idea is interesting, I think the paper needs a lot of work both in theory/simulation/diversity of tasks, and experimental tests.

**Strengths:**

The paper is well presented in general. The main idea behind it, both broadly, i.e. bridging machine learning and neuroscience, and specifically, i.e. the role of cognitive maps and value representations in decision making is important and the question of many researchers.

**Weaknesses:**

Most importantly, the evidence presented in favor of the claim is very limited, i.e., a simulation of one relatively simple task. Also, since many algorithms/network architectures can solve the task, it is essential to rule out other possibilities. It is also not clear to me why specific decisions were made in training the network and whether they play a key role in seeing the results or not.

**Questions:**

1) Testing these on real data would improve the contributions significantly.
2) How does the method work on other domains? For example, navigation tasks/mazes are very common in studying planning and the role of the hippocampus, especially in rodents. Does the network generate border cells or place cells related to the goal or informative areas?
3) How does the training process affect the result? If they do affect them significantly, what is the justification behind these theories (e.g., similarity to how animals learn)?
4) Multiple works have shown why showing the similarity of a network to representations in the brain is not enough to conclude these networks are implemented in the brain, e.g., "No Free Lunch from Deep Learning in Neuroscience: A Case Study through Models of the
Entorhinal-Hippocampal Circuit" and "What can 1.8 billion regressions tell us about the pressures shaping high-level visual representation in brains and machines?". How can the others show that their model is meaningful considering these works?

---

### Note · Authors · 2025-12-05

I have read and agree with the venue's withdrawal policy on behalf of myself and my co-authors.